# Blood Glucose Response of a Low-Carbohydrate Oral Nutritional Supplement with Isomaltulose and Soluble Dietary Fiber in Individuals with Prediabetes: A Randomized, Single-Blind Crossover Trial

**DOI:** 10.3390/nu14122386

**Published:** 2022-06-09

**Authors:** Eri Kokubo, Shunsuke Morita, Hirotaka Nagashima, Kazutaka Oshio, Hiroshi Iwamoto, Kazuhiro Miyaji

**Affiliations:** 1Health Care & Nutritional Science Institute, R&D Division, Morinaga Milk Industry Co., Ltd., 5-1-83 Higashihara, Zama, Kanagawa 252-8583, Japan; s-morita@morinagamilk.co.jp (S.M.); h_iwamot@morinagamilk.co.jp (H.I.); k_miyazi@morinagamilk.co.jp (K.M.); 2Medical Corporation Chiseikai Tokyo Center Clinic, 1-1-8 Yaesu, Chuou-ku, Tokyo 192-0397, Japan; nagashima_hirotaka@tc-clinic.jp; 3R&D Planning Department, R&D Division, Morinaga Milk Industry Co., Ltd., 5-1-83 Higashihara, Zama, Kanagawa 252-8583, Japan; k-ooshio@morinagamilk.co.jp

**Keywords:** glucose, insulin, oral nutritional supplement, prediabetes, isomaltulose, dietary fiber

## Abstract

A high-energy-type oral dietary supplement (ONS), with a low proportion of available carbohydrate (LC-ONS), which contains a slowly digestible carbohydrate, isomaltulose, and is fortified with soluble dietary fiber, was newly developed for individuals with diabetes or prediabetes. This study aimed to evaluate the impact of LC-ONS on blood glucose levels after ingestion in individuals with prediabetes. A single-blind, randomized crossover clinical trial was performed on 20 individuals with prediabetes. After overnight fasting, all subjects ingested one serving (200 kcal/125 mL) of either LC-ONS (40% energy proportion of available carbohydrates) or standard ONS (ST-ONS, 54% energy proportion of available carbohydrates) on two separate days. The incremental area under the curve of blood glucose levels for 120 min was significantly lower after LC-ONS ingestion compared to ST-ONS (2207 ± 391 mg/dL·min (least mean square value ± standard error) and 3735 ± 391 mg/dL·min, respectively; *p* < 0.001). The LC-ONS showed significantly lower blood glucose levels than the ST-ONS at all time points, except at baseline. Similarly, the incremental area under the curve of plasma insulin was significantly lower after LC-ONS ingestion. These results suggest that LC-ONS is useful as an ONS for energy supply in individuals with postprandial hyperglycemia.

## 1. Introduction

Older people are at an increased risk of malnutrition owing to decreased activity, loss of appetite, and diseases [1]. Oral nutritional supplements (ONS) are ready-to-use products that provide energy and balance macronutrients and micronutrients in a single serving. ONS were shown to improve the nutritional status of malnourished older individuals [2].

Conversely, diabetes is highly prevalent in older people [3]. In developed countries, the prevalence of diabetes [4,5] and prediabetes [6,7] is higher in older people than in younger people. ONS may lead to high postprandial blood glucose levels in people with diabetes and prediabetes because they are generally rich in rapidly digestible carbohydrates, such as starch hydrolysate. Postprandial hyperglycemia is considered a risk factor for the onset of cardiovascular disease [8] and microvascular complications [9] in patients with diabetes. Therefore, ONS for patients with diabetes and prediabetes should be designed to suppress postprandial blood glucose elevation.

There are three main strategies to suppress the postprandial blood glucose rise caused by ONS: (i) to reduce the energy proportion of available carbohydrates (AC, carbohydrates that are absorbed in the small intestine and increase blood glucose) and replace the energy with fat or protein [10,11]; (ii) using slowly digestible, low glycemic response-carbohydrates, such as isomaltulose and lactose as an AC source [12,13,14]; (iii) to increase the amount of soluble dietary fiber to slow down the rate of sugar absorption in the small intestine. To date, several ONS were developed that use these strategies alone or in combination and were reported to have lower postprandial blood glucose levels than standard ONS [10,11,12,13,14,15]. However, these products still have insufficient reports on the postprandial blood glucose levels; therefore, more evidence is required.

We have developed a new high-energy, low-carbohydrate-type ONS (LC-ONS), considering its use in patients with diabetes and prediabetes. LC-ONS has a low energy proportion of AC, which contains isomaltulose as a part of it. Furthermore, LC-ONS was fortified with soluble dietary fiber. This study aimed to evaluate the effect of LC-ONS on blood glucose levels after ingestion in individuals with prediabetes, compared to standard ONS.

## 2. Materials and Methods

### 2.1. Trial Design and Ethics

A single-center, randomized, single-blind, crossover clinical trial was conducted with 20 adult Japanese volunteers with prediabetes. This study evaluated the differences between the two types of ONS in blood glucose and insulin responses for 2 h after single oral ingestion. The primary outcome was the incremental area under the curve (iAUC) of the postprandial blood glucose responses. The secondary outcomes were the concentrations of blood glucose and insulin at each time point, the maximum value of blood glucose and insulin (Cmax), incremental value (iCmax) from baseline, and the iAUC of postprandial insulin. The study was conducted at the Medical Corporation Chiseikai Tokyo Center Clinic (Tokyo, Japan) between February and August 2021, in accordance with the principles of the Declaration of Helsinki (2013). The study protocol was approved by the Institutional Review Board of the Japan Conference of Clinical Research (Tokyo, Japan, protocol code, DMC1-01; date of approval, 19 February 2021). The study aims were carefully explained to all subjects, and written informed consent was provided. The protocol for this study was registered with the University Hospital Medical Information Network Clinical Trials Registry (No. UMIN000043590).

### 2.2. Subjects

From March 2021 to April 2021, individuals who were aware of prediabetes were recruited from the volunteer bank of a clinical research organization (3H Clinical Trial Inc., Tokyo, Japan). The candidates (*n* = 65) were judged to meet the following eligibility criteria in a screening test, including an oral glucose tolerance test using 75 g glucose (75 g OGTT). The inclusion criteria were age 20–64 years and prediabetic status. Based on the standards proposed by the Japan Diabetes Society [16,17], prediabetes was defined as meeting any of the following criteria in the screening test: (i) fasting plasma glucose (FPG) level of 100–125 mg/dL; (ii) 1-h glucose level of ≥180 mg/dL in 75 g OGTT; (iii) 2-h glucose level of 140–199 mg/dL in the 75 g OGTT; (iv) HbA1c value of 5.6–6.4%. Subjects who were already diagnosed with diabetes or who were likely to have diabetes from the results of the screening test by meeting both of the following criteria were excluded: (i) HbA1c value of ≥6.5%; (ii) FPG level of ≥126 mg/dL or 2-h glucose level of ≥200 mg/dL in the 75 g OGTT [17]. Other exclusion criteria were: milk and/or soybean allergy; collection of blood components or ≥200 mL of blood in the past 1 month; collection of ≥400 mL of blood in the past 4 months; use of medicines that could affect blood glucose levels; participation in another study; medical treatments or serious medical histories of diseases in the liver, kidneys, heart, lungs, digestive system, blood, endocrine, and metabolic systems; and those judged by the investigator to be inappropriate for the study. Eligible individuals who exceeded the target number of cases were also excluded.

During the screening test, information about subjects’ allergies, blood donation history for the past 4 months, medical history, smoking habits, and drinking habits was obtained using a questionnaire. Anthropometric measurements were performed. Fasting blood samples were collected to measure FPG, fasting insulin, HbA1c, triglycerides, total cholesterol, urea nitrogen, creatinine, aspartate aminotransferase (AST), alanine aminotransferase (ALT), δ-glutamyl transpeptidase (δ-GTP), red blood cells, white blood cells, and platelets. After fasting blood collection, subjects ingested a sugar solution containing 75 g of glucose (Toreran G solution 75 g, Yoshindo Inc., Toyama, Japan) for the 75 g OGTT, and blood was collected 1 and 2 h after ingestion to measure blood glucose levels. Twenty subjects were determined from the screening test results and randomly assigned to two groups (group A or group B). An allocation manager, who was independent of the trial staff, created the allocation order using the replacement block method (block size 4). The allocation ratio was set at 1:1. The subjects, but not the investigators, were blinded to the assignment.

### 2.3. Test Products

LC-ONS was prepared by Morinaga Milk Industry Co., Ltd. (Tokyo, Japan). A commercially available standard-type ONS (ST-ONS; Enjoy Climeal^®^; Clinico Co., Ltd., Tokyo, Japan) was used as control. The nutritional compositions of the two products are shown in Table 1. Both products are high-energy-type (1.6 kcal/mL) ONS with a serving size of 125 mL. LC-ONS contained isomaltulose (38.3% of AC), and had a lower content of AC than ST-ONS. LC-ONS also contains higher amounts of fat and dietary fiber than the ST-ONS.

### 2.4. Intervention

The study included two test days and a two-week washout period. On the first day, subjects in group A ingested ST-ONS, whereas subjects in group B ingested LC-ONS. After the washout, each subject ingested alternate products on the second day. The subjects were instructed not to eat or drink anything other than water after 21:00 on the day before the test. Consumption of alcohol was restricted to the day prior to the test. The subjects attended at 9:00 on the test day and were instructed to sit and rest until the end of the study. After a fasting blood collection at 10:00, the subjects received and ingested one serving (125 mL in a cup) of either LC-ONS or ST-ONS. Blood samples were collected at 15, 30, 45, 60, 90, and 120 min after ingestion to measure the levels of blood glucose and insulin. Adverse effects that occurred during the two weeks following ingestion of each product were collected through questionnaires or email interviews.

### 2.5. Laboratory Measurements

Blood glucose, insulin, HbA1c, triglyceride, total cholesterol, urea nitrogen, creatinine, AST, ALT, δ-GTP, red blood cells, white blood cells, and platelets were analyzed by a clinical testing laboratory (BML Inc., Tokyo, Japan). The iAUC values for blood glucose and insulin were calculated individually using the trapezoid rule, with the area beneath the baseline concentration omitted. Individual maximum values of blood glucose and insulin (Cmax) and the incremental values (iCmax), obtained by subtracting each baseline value from Cmax, were also determined.

### 2.6. Statistical Analyses

The sample size was set to 20, with reference to previous studies using other diabetes-specialized ONS products [10,11,13,14,15]. The baseline was calculated as the mean ± standard deviation or frequency. Primary and secondary outcomes and postprandial blood glucose and insulin levels were analyzed, using a linear mixed model. To analyze the iAUC and iCmax, the test products and test day were defined as fixed effects, and the subject ID was defined as a random effect. To analyze the Cmax and the concentrations at each time point, the test products, test day, and baseline value were defined as fixed effects, and the subject ID was defined as a random effect. The least mean square value and standard error for each group, the difference between groups, and the associated 95% confidence interval and *p*-value were calculated. The two-tailed significance level was set at 5%. The carryover effect was assessed by adding the test food-by-test day interactions to the main analysis model. Differences in the frequency of adverse events between the ingestion of LC-ONS and ST-ONS were evaluated using McNemar’s test. Statistical analyses were performed using JMP 13.2.1 (SAS Institute Inc., Cary, NC, USA).

## 3. Results

The flow of subjects throughout the study is shown in Figure 1. Sixty-five subjects were screened, and 20 subjects were enrolled and randomized into group A (*n* = 10) or group B (*n* = 10). All enrolled subjects completed the study. Table 2 shows the characteristics of the enrolled subjects’ age, sex, body mass index, and glycemic parameters. Triglyceride (100.8 ± 43.2 mg/dL), total cholesterol (214.1 ± 31.9 mg/dL), urea nitrogen (13.3 ± 3.3 mg/dL), creatinine (0.73 ± 0.16 mg/dL), AST (23.6 ± 7.4 U/L), ALT (20.9 ± 9.4 U/L), δ-GTP (28.3 ± 15.3 U/L), red blood cells (482.8 ± 42.2 × 10^4^/μL), white blood cells (5555 ± 1488 /μL), and platelets (26.6 ± 6.0 × 10^4^/μL) of subjects were in the healthy range.

### 3.1. Blood Glucose Responses

The iAUC of the blood glucose level was significantly lower after ingestion of the LC-ONS than after ingestion of the ST-ONS (Table 3). The blood glucose iAUC of the LC-ONS was 41% lower than that of the ST-ONS. The Cmax and iCmax of the blood glucose were also significantly lower after ingestion of LC-ONS than after ingestion of ST-ONS. Postprandial blood glucose levels at each time point after ingestion of the two products are shown in Figure 2. The LC-ONS showed significantly lower postprandial blood glucose levels than ST-ONS at all time points, except at baseline. No significant interactions were observed between the test products and test days (data not shown). This result suggested that carryover effects were not observed.

### 3.2. Insulin Responses

The iAUC of plasma insulin was significantly lower after ingestion of the LC-ONS compared to the ST-ONS ingestion (Table 3). The Cmax and iCmax of plasma insulin were also significantly lower after ingestion of LC-ONS. Postprandial plasma insulin levels at each time point after the ingestion of the two products are shown in Figure 3. The LC-ONS group showed significantly lower postprandial plasma insulin levels than the ST-ONS group at 60, 90, and 120 min after consumption. 

### 3.3. Safety

Five subjects reported seven adverse events during the study. Three events reported in the LC-ONS period were studied, and four events reported when in the ST-ONS period were studied. There were no significant differences in the incidence between the LC-ONS and ST-ONS ingestion periods (*p* = 0.654). One minor adverse event (mild diarrhea for one day after ingestion) was found to be a side effect. This event was not a safety issue and was predicted in advance because LC-ONS contains lactulose and inulin, which may cause gastrointestinal symptoms. The other six adverse events (gastrointestinal disorders and headache) were not considered to be related to the intake of the test products.

## 4. Discussion

Our results showed that the LC-ONS caused a lower increase in blood glucose levels after consumption than the standard-type ONS. Diets with a low AC energy ratio have a low postprandial blood glucose rise [18]. Isomaltulose, a disaccharide consisting of glucose and fructose linked by α-1,6 glycosidic linkages, is a low glycemic response carbohydrate [19,20]. Isomaltulose is an AC that is completely digested and absorbed in the small intestine, but its digestion rate is slower than that of sucrose or maltose. Resistant maltodextrin and inulin contained in the LC-ONS are soluble dietary fibers that inhibit the absorption of carbohydrates ingested simultaneously and suppress postprandial blood glucose elevation [21,22]. The results of the LC-ONS may be attributed to the combined effects of these three elements: the adjustment of the “quantity” and “quality” of AC, and the enrichment of soluble dietary fiber.

ONS with low amounts of AC or high percentages of carbohydrates with a low glycemic response in the AC have significantly lower postprandial blood glucose elevation. The ONS with half of the AC showed more than 60% lower iCmax than the standard ONS in postprandial blood glucose [11]. The ONS with 56% of the carbohydrate replaced with isomaltulose showed a 55% lower postprandial blood glucose iAUC than the standard ONS with similar proportions of macronutrients [12]. In contrast, the ONS with 36% of the AC replaced with isomaltulose resulted in only a 15% reduction in postprandial blood glucose iAUC compared to a standard ONS with the same amount of AC [14]. Thus, it is estimated that more than half of the AC might be reduced or replaced with a low glycemic response carbohydrate, to achieve a significant reduction in blood glucose elevation. However, an extreme reduction in the amount of AC would represent a significant departure from the composition of each country’s dietary guidelines [23,24,25]. Increased intake of isomaltulose as a carbohydrate source for ONS leads to increased intake of its constituent sugar, fructose. Chronic intake of fructose and sucrose is associated with the development of the metabolic syndrome and obesity [26,27]. Conversely, isomaltulose may have a different effect on metabolism than fructose and sucrose, probably because of the slower entry rate of fructose into the blood or the lower postprandial blood glucose rise. In patients with type 2 diabetes, the continuous daily consumption of 50 g of isomaltulose instead of sucrose was reported to decrease blood triglyceride levels [28]. Rats continuously fed with isomaltulose suppressed the increase in abdominal fat accumulation [29]. However, the effects of the continuous ingestion of large amounts of isomaltulose in humans require further evaluation.

As for the effect of soluble dietary fiber, the ONS with 23% of carbohydrates replaced with tapioca-derived resistant maltodextrin showed about 5% lower peak value of postprandial blood glucose than the standard ONS, but the difference was not significant [30]. On the other hand, the ONS with 8% of total carbohydrates replaced with soluble dietary fiber (polydextrose) and 15% with a low glycemic response-carbohydrate (fructose) reported a 33% reduction in postprandial blood glucose iAUC compared to the control ONS [31]. This report suggests that the combination of soluble dietary fibers and low glycemic response carbohydrates may have a combined effect. The LC-ONS in this study was adjusted for three types of adjustments: a 25% reduction in AC content compared to ST-ONS; replacement of 38% of the AC with isomaltulose; and increasing soluble dietary fiber by 3.5 g than the ST-ONS, resulting in a 41% reduction in the iAUC of postprandial blood glucose.

The ONS for postprandial blood glucose control is expected to suppress insulin secretion in addition to postprandial blood glucose levels by reducing the amount and rate of carbohydrate influx from the small intestine into the bloodstream. Suppression of postprandial insulin secretion was observed with other ONS for diabetes [10,15,31] and low-glycemic index diets [32]. As predicted, LC-ONS also showed low postprandial insulin levels.

The limitations of this study were, first, that only the combined effect was examined, and the contribution of each element and the presence of synergistic effects are unknown. To clarify this, a comparison of the effects of each element alone and in combination is required. Second, it is difficult to assess the clinical benefit by comparing the glycemic response after a single dose with the standard ONS. Although many prior studies have used commercially available, commonly used ONS as the standard [10,11,12,13,14,15], their nutritional composition was not consistent. To assess the clinical usefulness of LC-ONS, it is desirable to compare it with products that are clinically useful in patients with diabetes or prediabetes [33,34]. It is also necessary to use indices that are used to make clinical decisions about glycemic control in patients with diabetes, such as the effect on daily blood glucose variability in continuous glucose monitoring and HbA1c levels after continuous intake. Third, the effect on diabetic patients was unknown because the subjects of this study were limited to prediabetic patients and did not include diabetic patients. The results of LC-ONS found in this study are presumed to be found in patients with diabetes as well because the postprandial blood glucose-lowering effects of isomaltulose [35] and soluble dietary fibers [36,37] were found in patients with diabetes in previous studies. However, studies in patients with diabetes are desirable to clarify whether LC-ONS is effective even in patients on medication or with severe postprandial hyperglycemia.

As a substitute for AC energy sources, LC-ONS has a fat energy ratio of 40%, which is higher than the dietary guidelines of many countries [23,24,25]. However, medium-chain fatty acids, which comprise 45% of the constituent fatty acids of LC-ONS, are known to be less likely than long-chain fatty acids to accumulate ectopic fat [38] and to increase insulin sensitivity [39]. Therefore, LC-ONS may have few adverse effects on lipid metabolism and insulin sensitivity, even with its high fat composition. Furthermore, soluble dietary fiber was reported to ameliorate insulin resistance caused by chronic inflammation by improving dysbiosis of the intestinal microflora in continuous ingestion [40,41], as well as their postprandial blood glucose-lowering effects in one-shot ingestion. Therefore, the LC-ONS may improve glycemic control by continuously ingesting large amounts of soluble dietary fiber; however, the effect of continuous ingestion is unclear. It is desirable to evaluate the effect of continuous ingestion on glycemic control, such as HbA1c levels, and the effect on intestinal microbiota.

## 5. Conclusions

LC-ONS showed lower postprandial blood glucose compared to the standard type of ONS; additionally, insulin response was low. These results suggest that LC-ONS is useful as a high-energy-type ONS for energy supply in people with diabetes and prediabetes.

## Figures and Tables

**Figure 1 nutrients-14-02386-f001:**
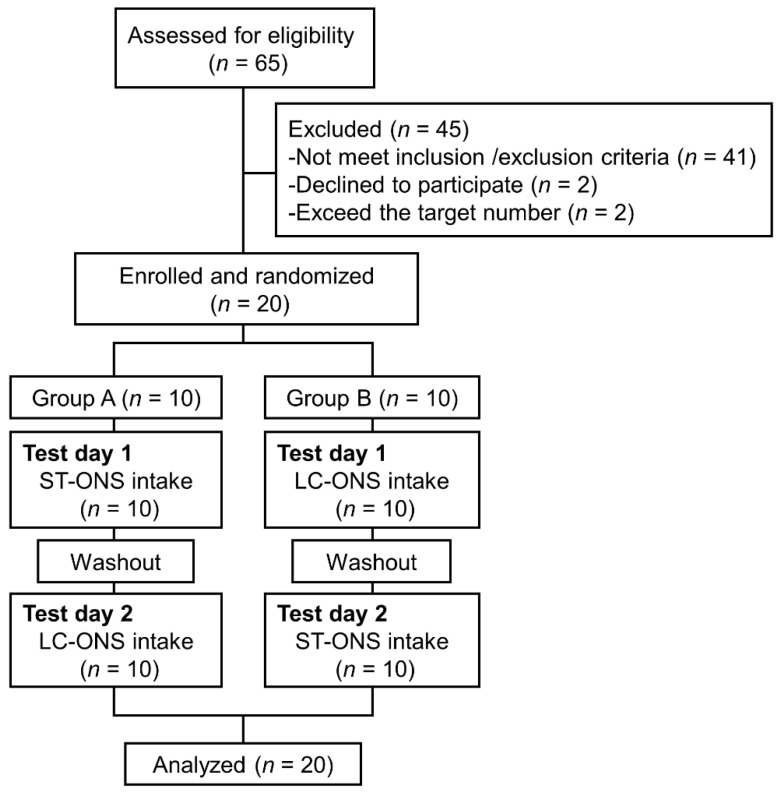
Flow chart of the study.

**Figure 2 nutrients-14-02386-f002:**
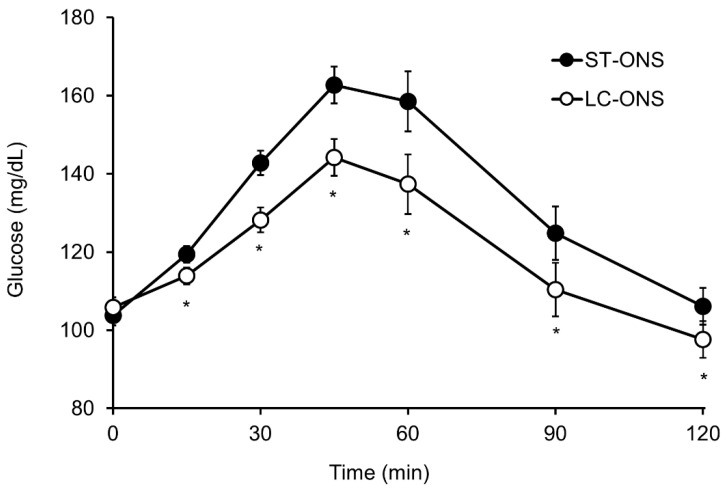
Postprandial levels of glucose after ingestion of a low-carbohydrate-type oral nutritional supplement (LC-ONS) or a standard-type ONS (ST-ONS) (*n* = 20). Values are expressed as the least mean square value ± standard error; * *p* < 0.05 comparing ST-ONS vs. LC-ONS.

**Figure 3 nutrients-14-02386-f003:**
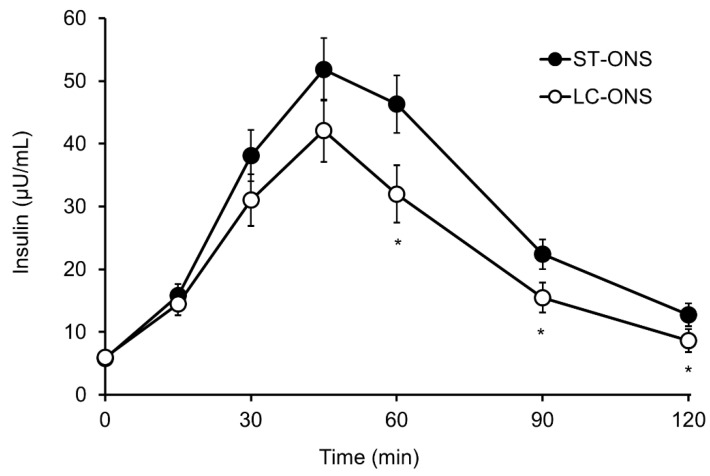
Postprandial levels of insulin after ingestion of a low-carbohydrate-type oral nutritional supplement (LC-ONS) or a standard-type ONS (ST-ONS) (*n* = 20). Values are expressed as the least mean square value ± standard error; * *p* < 0.05 comparing ST-ONS vs. LC-ONS.

**Table 1 nutrients-14-02386-t001:** Composition of the test products.

	LC-ONS (125 mL)	ST-ONS (125 mL)
Energy kcal	200	200
Protein g (En%) ^1^	7.5 (15)	7.5 (15)
Fat g (En%)	8.9 (40)	6.7 (30)
Carbohydrate g (En%)	27.1 (45)	29.3 (55)
Available carbohydrate g (En%)	20.1 (40)	26.8 (54)
Isomaltulose g	7.7	0
Indigestible carbohydrate g (En%)	7.0 (5)	2.5 (1)
Lactulose g	1.0	0
Dietary fiber g	6.0	2.5
Resistant maltodextrin g	4.5	2.5
Inulin g	1.5	0

^1^ En% = Energy ratio.

**Table 2 nutrients-14-02386-t002:** Subject characteristics at the screening visit (*n* = 20). The data are expressed as mean ± standard deviation or frequency.

Parameter	Value ^1^
Age (years)	54.0 ± 5.6
Sex (Male/Female)	11/9
Body mass index (kg/m^2^)	23.1 ± 3.4
HbA1c (%)	5.69 ± 0.32
Fasting blood glucose (mg/dL)	106.0 ± 10.7
Glucose at 1 hr (mg/dL)	195.3 ± 31.7
Glucose at 2 hr (mg/dL)	153.8 ± 48.4
Fasting plasma insulin (µU/mL)	6.36 ± 3.62

^1^ The data are expressed as mean ± standard deviation or frequency.

**Table 3 nutrients-14-02386-t003:** Area under curve (AUC), maximal value (Cmax), and incremental maximal value (iCmax) of glucose and insulin after consumption of the test products.

		ST-ONS (*n* = 20) ^1^	LC-ONS (*n* = 20) ^1^	Difference (95% CI) ^2^	*p*-Value
Glucose	iAUC (mg/dL∙min)	3735 ± 391	2207 ± 391	−1528 (−2150 to −905)	<0.001
	Cmax (mg/dL)	173.9 ± 4.9	149.5 ± 4.9	−24.4 (−33.7 to −15.2)	<0.001
	iCmax (mg/dL)	68.6 ± 4.9	45.2 ± 4.9	−23.5 (−32.8 to −14.1)	<0.001
Insulin	iAUC (µU/mL∙min)	2814 ± 317	2007 ± 317	−807 (−1251 to −363)	0.001
	Cmax (µU/mL)	61.9 ± 5.0	45.4 ± 5.0	−16.6 (−26.9 to −6.2)	0.004
	iCmax (µU/mL)	55.6 ± 7.0	40.0 ± 7.0	−15.6 (−24.8 to −6.3)	0.002

^1^ Values are expressed as the least mean square value ± standard error; ^2^ CI = confidence interval.

## Data Availability

The data presented in this study are available on request from the corresponding author.

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
