# Peer review of "Blood Glucose Response of a Low-Carbohydrate Oral Nutritional Supplement with Isomaltulose and Soluble Dietary Fiber in Individuals with Prediabetes: A Randomized, Single-Blind Crossover Trial"

_nutrients, 2022, doi:10.3390/nu14122386_

Round 1

Reviewer 1 Report

Paper entitled "Blood Glucose Response of a Low-Carbohydrate Oral Nutritional Supplement with Isomaltulose and Soluble Dietary Fiber in Individuals with Prediabetes: A Randomized, Single-Blind Crossover Trial" is dealing with solving the problem of malnutrition in the elderly with diabetes. The methods used are properly described and appropriate for this type of research. Results are clearly presented, and conclusions are supported by the results. I am not sure how interesting this topic is to the readers.

Author Response

Dear Reviewer 1

Thank you very much for providing important comments. Below are our responses to your comments.

Reviewers' comments

Paper entitled "Blood Glucose Response of a Low-Carbohydrate Oral Nutritional Supplement with Isomaltulose and Soluble Dietary Fiber in Individuals with Prediabetes: A Randomized, Single-Blind Crossover Trial" is dealing with solving the problem of malnutrition in the elderly with diabetes. The methods used are properly described and appropriate for this type of research. Results are clearly presented, and conclusions are supported by the results. I am not sure how interesting this topic is to the readers.

Reply

Thank you for your comment that the methods used are properly described and appropriate. We also appreciate your comments that the results are clearly presented and that the conclusions are supported by the results.

LC-ONS is a practical, yet novel composition of ONS that combines well-known blood glucose elevation control strategies. We believe that our report showing a 41% reduction in postprandial blood glucose elevation with this composition will be an interesting topic for readers of Nutrients, including physicians and nutritionists facing the problem of malnutrition in older people with diabetes.

Again, we are thankful for the time and power you expended.

Sincerely,

Eri Kokubo
Health Care & Nutritional Science Institute,
R&D Division, Morinaga Milk Industry Co., Ltd.
5-1-83 Higashihara, Zama, Kanagawa 252-8583, Japan
Tel: +81-46-252-3057, Fax: +81-46-252-3055
E-mail address: [email protected]

Reviewer 2 Report

In this article the authors explore effects of a new type of dietary supplement that has a low proportion of available carbohydrates on blood glucose and insulin levels in prediabetes patients, compared to standard dietary supplement.

The article is well written, it is easy to understand. The language used follows the norms of scientific English language. I believe that the methods and controls used are sufficient for the conclusions that are stated at the end of the article. There are limitations and shortcomings of the work, but they are clearly stated and explained.

I only have one suggestion for the improvement of the article.

L292 and L293, please consider changing the statement of insulin and glucose levels being „suppressed“. My understanding is that your product does not suppress blood glucose levels and insulin, it only makes them not rise as much, so please consider choosing another word, like „lower“

Author Response

Dear Reviewer 2

Thank you very much for providing important comments. Below are our responses to your comments.

Reviewers' comment 1

In this article the authors explore effects of a new type of dietary supplement that has a low proportion of available carbohydrates on blood glucose and insulin levels in prediabetes patients, compared to standard dietary supplement.

The article is well written, it is easy to understand. The language used follows the norms of scientific English language. I believe that the methods and controls used are sufficient for the conclusions that are stated at the end of the article. There are limitations and shortcomings of the work, but they are clearly stated and explained.

Reply to comment 1

Thank you for your comment that our article is well written and easy to understand. We also appreciate your comment that the methods and controls used are sufficient for the conclusions that are stated at the end of the article.

Reviewers' comment 2

I only have one suggestion for the improvement of the article.

L292 and L293, please consider changing the statement of insulin and glucose levels being „suppressed“. My understanding is that your product does not suppress blood glucose levels and insulin, it only makes them not rise as much, so please consider choosing another word, like „lower“

Reply to comment 2

Thank you for pointing this out. We have changed the following sentence:

From: (L.292 – L.293 in the first submitted version)

“LC-ONS showed suppressed postprandial blood glucose compared to the standard type of ONS; additionally, insulin response was suppressed.” 

To: (L.293 – L.294 in the revised version)

“LC-ONS showed lower postprandial blood glucose compared to the standard type of ONS; additionally, insulin response was low.”

Other: From the “Review Report Form”

Are all the cited references relevant to the research? – “Can be improved”

Reply

Our manuscript had some difficulty to judge if all the references cited were relevant to the study because of the inclusion of references in only Japanese and Chinese. To improve this point, we have changed two references cited in the manuscript, and the corresponding text as follows:

[Reference in Japanese] 

Reference [22] in the first submitted version

Taguchi-Yanagisawa, C.; Togashi, H.; Kondo, K. Effect of a Concentrated Liquid Diet Fortified with Resistant Maltodextrin on Postprandial Blood Glucose - Single Dose Study (in Japanese). Risnyo Eiyo 2008, 113, 905–909.

This reference has been deleted.  Instead, we have substituted the following for reference [30], a paper evaluating the effect of soluble fiber on blood glucose elevation in the ONS.

Reference [30] in the revised version

Astina, J.; Saphyakhajorn, W.; Borompichaichartkul, C.; Sapwarobol, S. Tapioca Resistant Maltodextrin as a Carbohydrate Source of Oral Nutrition Supplement (ONS) on Metabolic Indicators: A Clinical Trial. Nutrients 2022, 14, 916, doi:10.3390/nu14050916.

Changes to the corresponding text:

From: L.215 – L.218 in the First submitted version

Resistant maltodextrin and inulin contained in LC-ONS are soluble dietary fibers that in-hibit the absorption of carbohydrates ingested simultaneously and suppress postprandial blood glucose elevation [21–23].

To: L.215 – L.218 in the revised version

Resistant maltodextrin and inulin contained in LC-ONS are soluble dietary fibers that in-hibit the absorption of carbohydrates ingested simultaneously and suppress postprandial blood glucose elevation [21,22]. 

From: L.243 – L.248 in the First submitted version

As for the effect of soluble dietary fiber, ONS with 4.4 g per 200 kcal of resistant maltodextrin was reported to have approximately 25% lower blood glucose iAUC than ONS with the same macronutrient ratio but no fiber [22] Moreover, ONS with 8% of total carbohydrates replaced with soluble dietary fiber (poly-dextrose) and 15% with a low glycemic response-carbohydrate (fructose) reported a 33% reduction in postprandial blood glucose iAUC compared to control ONS [31].

To: L.243 – L.249 in the revised version

As for the effect of soluble dietary fiber, ONS with 23% of carbohydrates replaced with tapioca-derived resistant maltodextrin showed about 5% lower peak value of postprandial blood glucose than the standard ONS, but the difference was not significant [30]. On the other hand, ONS with 8% of total carbohydrates replaced with soluble dietary fiber (polydextrose) and 15% with a low glycemic response-carbohydrate (fructose) reported a 33% reduction in postprandial blood glucose iAUC compared to control ONS [31]. 

[Reference in Chinese] 

Reference [24] in the first submitted version

Chinese Nutrition Society. Chinese Dietary Reference Intakes 2013 (in Chinese); China Light Industry Publishing House Beijing, China, 2013

As example of each country's dietary guidelines, we replaced this Chinese Dietary Reference Intakes to an English-language citation, Food Based Dietary Guidelines in the WHO European Region.

Reference [23] in the revised version

World Health Organization. Food Based Dietary Guidelines in the WHO European Region; Copenhagen, Denmark, 2003; Vol. 79832;.

Changes to the corresponding text: 

No modifications have been made to the corresponding text (L.231 – L.232 and L.278 – L.279 in the first submitted version, L.231 – L.232 and L.279 – L.280 in the revised version).

Again, thank you for giving us the opportunity to strengthen our manuscript with your valuable comments and queries. We hope that these revisions persuade you to accept our submission.

Sincerely,

Eri Kokubo
Health Care & Nutritional Science Institute,
R&D Division, Morinaga Milk Industry Co., Ltd.
5-1-83 Higashihara, Zama, Kanagawa 252-8583, Japan
Tel: +81-46-252-3057, Fax: +81-46-252-3055
E-mail address: [email protected]
